# Different Transcutaneous Auricular Vagus Nerve Stimulation Parameters Modulate the Anti-Inflammatory Effects on Lipopolysaccharide-Induced Acute Inflammation in Mice

**DOI:** 10.3390/biomedicines10020247

**Published:** 2022-01-24

**Authors:** Yoon-Young Go, Won-Min Ju, Chan-Mi Lee, Sung-Won Chae, Jae-Jun Song

**Affiliations:** 1Department of Otorhinolaryngology-Head and Neck Surgery, Korea University Guro Hospital, 80 Guro-dong, Guro-gu, Seoul 08308, Korea; gokogoko@korea.ac.kr (Y.-Y.G.); juwonmin@korea.ac.kr (W.-M.J.); cksal7873@korea.ac.kr (C.-M.L.); chaeorl@korea.ac.kr (S.-W.C.); 2Institute for Health Care Convergence Center, Korea University Guro Hospital, Seoul 08308, Korea; 3Neurive Institute, Neurive Co., Ltd., Seoul 08308, Korea

**Keywords:** transcutaneous auricular vagus nerve stimulation (taVNS), anti-inflammation, cytokines, electrical stimulation parameters, coronavirus disease 2019 (COVID-19)

## Abstract

Vagus nerve stimulation (VNS) is considered a potential method for anti-inflammation due to the involvement of the VN in the cholinergic anti-inflammatory pathway (CAP) formation of a connection between the central nervous system and peripheral immune cells that help relieve inflammation. However, whether a non-invasive transcutaneous auricular VNS (taVNS) modulates the inflammation levels via altering the parameter of taVNS is poorly understood. This study aimed to determine the differential inhibitory effects of taVNS on lipopolysaccharide (LPS)-induced systemic inflammation using electrical stimulation parameters such as pulse frequency and time. The taVNS-promoted CAP activity significantly recovered LPS-induced tissue injuries (lung, spleen, and intestine) and decreased inflammatory cytokine levels and tissue-infiltrated immune cells. Interestingly, the anti-inflammatory capacity of taVNS with 15 Hz was much higher than that of taVNS with 25 Hz. When a cytokine array was used to investigate the changes of inflammation and immune response-related cytokines/chemokines expression in taVNS with 15 Hz or 25 Hz treatment in LPS-induced endotoxemia in mice, most of the expression of cytokines/chemokines associated with pro-inflammation was severely decreased in taVNS with 15 Hz compared to 25 Hz. This study demonstrated that the taVNS parameter could differentially modulate the inflammation levels of animals, suggesting the importance of taVNS parameter selection for use in feasible interventions for acute inflammation treatment.

## 1. Introduction

The vagus nerve (VN) controls the parasympathetic nervous system as one of the cranial nerves [1]. Over 80% of afferent nerves in the VN mostly convey the body’s sensory information to the central nervous system [2]. For example, once the VN detects any inflammatory process in the body, the efferent fibers of the VN activate postsynaptic excitatory potentials to modulate immune response via the α-7 nicotinic acetylcholine receptors (α7nAChR)-mediated pathway [3,4]. Acetylcholine from vagal efferent fibers interacts with α7nAChR in the immune cells like macrophages and dendritic cells of tissues, blocking the release of pro-inflammatory cytokines, including tumor necrosis factor-alpha (TNF-α), interleukin-6 (IL-6), IL-1β, and IL-8 [5,6]. These inflammatory reflex reactions reduce cytokine production and inhibit the body’s systemic inflammatory response [5]. Considering the anti-inflammatory modulation of the VN, VN stimulation (VNS) is a medical tool for treating inflammatory-related diseases. Many researchers have evaluated the effect of electrical stimulation of the VN on inflammatory disorders such as lung injury, sepsis, and rheumatoid arthritis [5,7,8,9].

An electrical stimulator of the VN was approved by the United States Food and Drug Administration to treat epilepsy in 1997 [10]. Early devices of VNS delivered the electrical impulses to the VN invasively. An electrode was surgically implanted into the left chest under the collarbone and then connected to the left VN of the patient [11,12]. More recent VNS devices are non-invasive and easily modulate VNS via the transcutaneous region such as the auricular region [13,14]. VNS devices were classically generated to treat drug-resistant diseases such as epilepsy or depressive disorder [10,13,15]. VNS provides a therapeutic intervention to avoid the side effects of chemical agents as a non-drug therapy.

Currently, the world is witnessing the coronavirus disease 2019 (COVID-19) pandemic [16,17]. COVID-19 often causes extreme immune reactions in the human body, which may lead to severe organ damage via an inflammatory cytokine storm [18,19]. Thus, preventing or modulating cytokine release is an important strategy to impede multi-tissue damage-related mortality in COVID-19. To reduce the excessive inflammatory cytokine levels in COVID-19, targeting α7nAChR activity via VNS can effectively control the further aggravation induced by the activation of the immune system. Several researchers also suggested that VNS could be a potential adjunct therapy for inflammatory disorders originating from COVID-19 [20,21,22]. In this study, we used a lipopolysaccharide (LPS)-induced endotoxin mice model to upregulate the expression levels of inflammatory factors. The anti-inflammatory effects of transcutaneous auricular VNS (taVNS) were observed in the lung, spleen, and intestines, which are innervated by the α7nAChR-mediated cholinergic anti-inflammatory pathway (CAP) of the VN [5,6,23]. We also determined that different combinations of electrical parameters such as the frequency and time of taVNS affected the expression levels of pro- and/or anti-inflammatory cytokines in the lung, spleen, intestine, and serum. We found that specific stimulation parameters of taVNS can be used to modulate the rate of inflammation in vivo.

The present study reveals the effectiveness of taVNS in reducing systemic inflammation and demonstrates the therapeutic potential of taVNS in the treatment of acute inflammation, to be considered for application in clinical trials for COVID-19 as an adjuvant therapy.

## 2. Materials and Methods

### 2.1. LPS-Induced Endotoxemia Mice Model

This study used 6–8-week-old male C57BL/6 mice that weighed 25~27 g (purchased from Orientbio Inc., Seongnam, Korea). All animal experiments were approved by the Institutional Animal Care and Use Committee (IACUC) of Korea University and followed the animal ethics and welfare standards according to the IACUC guidelines. To prepare the LPS-induced endotoxemia model, mice were anesthetized by isoflurane. A single intraperitoneal injection of LPS (2 mg/kg, *Escherichia coli* 0111: B4; Sigma Aldrich, St. Louis, MO, USA) was conducted between stimulation with taVNS. Animals were sacrificed 2 h after LPS administration, based on previous studies [24,25] and our own preliminary analysis, and then serum and tissue samples were collected. All samples were stored at −80 °C until use.

### 2.2. Transcutaneous Auricular VNS

Two electrodes coated with a gold-plated platinum hook were placed on the auricular concha of the left ear (Figure 1A). Electrodes were connected to a stimulator (Neurive Inc., Seoul, Korea). Both the cymba and cavum conchae of the auricular were biphasically stimulated with the same parameter. The LPS was applied between two taVNS treatments with the indicated stimulation parameter in each group (Figure 1B,C). The charged-balanced biphasic waveforms during stimulation were shown in Figure 1D.

### 2.3. Administration of α7nAChR Antagonist

Methyllycaconitine citrate (MLA) (Tocris Bioscience, Bristol, Avon, UK) was used for the α7nAChR intervention as a specific α7nAChR antagonist according to a previous method [26]. MLA was dissolved in phosphate-buffered saline (PBS) and then administered to mice intraperitoneally at a dose of 5 mg/kg before LPS injection (Figure 1B).

### 2.4. Enzyme-Linked Immunosorbent Assay (ELISA)

Serum and intestine tissues from mice were used to determine the levels of pro-inflammatory cytokines. The concentrations of TNF-α, IL-6, and IL-1β were analyzed by mouse-specific ELISA kits (R&D Systems, Minneapolis, MN, USA), according to the manufacturer’s instructions.

### 2.5. Western Blotting

Total proteins of spleen tissues were lysed. An equal amount of protein was subjected to immunoblotting using antibodies against myeloperoxidase (MPO) (1:1000; Invitrogen, Carlsbad, CA, USA) and β-actin (1:2000; Santa Cruz Biotechnology, Dallas, TX, USA) for primary antibodies. The same amounts of proteins were electrophoresed on SDS-PAGE and then transferred onto PVDF membranes (Millipore, Darmstadt, Germany). Blocked membranes with 5% skim milk were incubated with primary antibodies overnight at 4 °C. The next day, the membranes were incubated with a secondary antibody (HRP-goat anti-rabbit IgG antibody, 1:3000, Invitrogen) for 1 h at room temperature. Images were subsequently captured using a Fusion Solo Imaging System (Vilber Lourmat, Marne-la-Vallée, France). Immunoreactive protein bands were quantified using ImageJ software.

### 2.6. Hematoxylin and Eosin Staining

At sacrifice, lung and intestine tissues were removed from the mice, fixed in 4% paraformaldehyde, embedded in paraffin, and sliced into 5-μm sections using a rotary microtome (Leica RM2255, Weztlar, Germany). For hematoxylin and eosin (H&E) staining, the deparaffinized tissue section was incubated in hematoxylin solution (Sigma) for 5 min and Eosin-Y solution (Sigma) for 1 min with tap water washing. Images were taken using Olympus BX43 microscope (Olympus Co., Tokyo, Japan). Inflammatory cell accumulation in the alveolar space, interalveolar septum thickening, alveolar congestion, alveolar hemorrhage, and cellular hyperplasia were considered for lung injury scoring: nil, 0; mild, 1; moderate, 2; severe, 3 [27]. Morphological examination with these five pathological features was performed in blind analysis by two examiners. The villus height was determined by the vertical distance from the crypt opening to the tip of the villus. The crypt depth was defined from the base of the level of the crypt opening.

### 2.7. Myeloperoxidase Staining

The deparaffinized 5-μm sections were incubated in MPO antibody (1:1000, Invitrogen) for 30 min at room temperature and washed with PBS. They were then incubated with a secondary antibody (1:1000, peroxidase-labeled goat anti-rabbit IgG) for 30 min. After the final wash with PBS, diaminobenzidine (DAB; DAKO, Santa Clara, CA, USA) was applied on the slide to detect the bound antibody. Hematoxylin was then used to evaluate the presence of neutrophils, as described previously [28].

### 2.8. RNA Extraction and Quantitative Real-Time Polymerase Chain Reaction

Total RNA was extracted from lung and spleen tissues using TRIzol™ reagent (Invitrogen) to analyze the relative gene expression. Then, PrimeSript™ first strand cDNA Synthesis Kit (Takara Bio, Tokyo, Japan) was used for the reverse transcription of 1 μg of RNA to 20 μL of cDNA, according to the manufacturer’s instructions. Polymerase chain reaction (PCR) was performed using the obtained cDNA as a template with the Power^®^ SYBR Green PCR Master Mix kit (Life Technologies Co. Ltd., Woolston, UK). The relative expression levels of *TNF-**α*, *IL-6*, *IL-1β*, *IL-8*, *TGF-β*, and *IL-10*, were calculated by the 2^(^^−∆∆Ct)^ method with normalization to *β-actin*. The specific primer sequences used in this study were: *TNF-**α*, 5′-CCC CAA AGG GAT GAG AAG TT-3′ (forward) and 5′-CAC TTG GTG GTT TGC TAC GA-3′ (reverse); *IL-6*, 5′-CCG GAG AGG AGA CTT CAC AG-3′ (forward) and 5′-CAG AAT TGC CAT TGC ACA AC-3′ (reverse); *IL-1**β*, 5′-TCG CAG CAG CAC ATC AAC AAG-3′ (forward) and 5′-CAT GTC CTC ATC CTG GAA G-3′ (reverse); *IL-8*, 5′-CCC GCG TTA GTC TGG TGT AT-3′ (forward) and 5′-AAC AGC CCA TAG TGG AGT GG-3′ (reverse); *TGF-**β*, 5′-TTG CTT CAG CTC CAC AGA GA-3′ (forward) and 5′-TGG TTG TAG AGG GCA AGG AC-3′ (reverse); *IL-10*, 5′-ATG CAG GAC TTT AAG GGT TAC TTG-3′ (forward) and 5′-AGA CAC CTT GGT CTT GGA GCT TA-3′ (reverse); *β-actin*, 5′-AGC CAT GTA CGT AGC CAT CC (forward) and 5′-CTC TCA GCT GTG GTG GTG AA-3′ (reverse).

### 2.9. Cytokine Array

To analyze the inflammatory-related cytokines, chemokines, growth factors, and angiogenic markers in serum samples, the concentration and purity of the isolated proteins were determined using the BCA protein assay kit (Pierce, Rockford, IL, USA) and UV spectrum. The antibody array slide (RayBiotech, Norcross, GA, USA, #L308) consisted of 308 nitrocellulose membrane kits to detect 308 mouse proteins in duplicated capture antibodies with positive and negative control antibodies. Briefly, the array slide was blocked with 400 μL of blocking solution for 30 min and incubated with samples for 2 h at room temperature. After being washed with the manufacturer-supplied buffers, the membranes were immersed in biotin-conjugated anti-cytokine antibodies and then incubated for 2 h with gentle shaking. Subsequently, Cy3-conjugated streptavidin solution was added to generate the chemiluminescent signals at each spot in the membrane. The fluorescence signal intensity was measured using GenePix 4100A microarray scanner (Axon Instrument, San Jose, CA, USA) within 24–48 h at 10-μm resolution, optimal laser power, and photomultiplier (PMT). The quantified scan images with GenePix software (Axon Instrument, San Jose, CA, USA) calculated the average signal of the duplicate spots and then normalized it to the control spot signals. The protein information for data mining was annotated using UniProt DB. Graphic visualization was used in ExDEGA software (Ebiogen Inc., Seoul, Korea).

### 2.10. Statistical Analysis

All data were obtained from triplicate experiments and have been expressed as the mean ± standard deviation. A student’s two-tailed *t*-test and one-way analysis of variance with Prism 5 software (GraphPad, San Diego, CA, USA) were used. The *p*-values are shown in the figures, and the differences were considered statistically significant at * *p* < 0.05, ** *p* < 0.01, and *** *p* < 0.001.

## 3. Results

### 3.1. taVNS Reduced the Expression Levels of Pro-Inflammatory Cytokines in the Serum of the LPS-Induced Inflammation

The pro-inflammatory cytokines were rapidly evoked in the LPS-induced endotoxemia group. After systemic inflammation via LPS administration, TNF-α and IL-1β highly increased in serum. Electrical stimulation with taVNS significantly inhibited the expression levels of TNF-α and IL-1β, indicating excellent anti-inflammatory efficacy of taVNS (Figure 2A). We also analyzed the CAP-mediated inhibitory effect of taVNS on systemic inflammation using LPS application pre-treatment with the α7nAChR antagonist, MLA. A reverse increase in TNF-α and IL-1β was detected in MLA-treated group with taVNS on LPS-induced inflammation compared to the MLA-untreated group (LPS + taVNS) (Figure 2A). These results indicate that the decrease in pro-inflammatory cytokines levels by taVNS was weakened by inhibiting α7nAChR using MLA. MLA treatment slightly promoted the levels of pro-inflammatory cytokines compared with LPS injection alone, but not significantly. These ELISA results demonstrated that taVNS reduced the expression levels of pro-inflammatory cytokines in LPS-induced systemic inflammation via activation of α7nAChR.

We found two studies that have reported that specific VNS parameters can affect the cytokine levels in serum [24,29]. It is unclear whether the parameters of taVNS, specifically pulse frequency and time, differentially affect the inflammatory cytokine expression in the serum of the endotoxin model. To address this, we delivered taVNS with 15 and 25 Hz for 5 and 10 min to LPS-induced endotoxemia individually (n = 5–10). Other parameters, including the pulse width and amplitude, were not changed. The electrical stimulation videos of taVNS with 15 Hz and 25 Hz were represented in Supplementary video, respectively. All experimental groups with taVNS in LPS-induced inflammation significantly decreased the serum TNF-α and IL-1β levels compared to the non-taVNS group in LPS induction (Figure 2B). In particular, taVNS stimulation with low pulse frequency (15 Hz) produced a significant inhibitory effect on serum TNF-α and IL-1β compared to the high pulse frequency (25 Hz) regardless of the time parameter. These results demonstrate that taVNS had anti-inflammatory effects via activation of α7nAChR on LPS endotoxemia, and pulse frequency of taVNS can be an important parameter for the regulation of pro-inflammatory cytokines levels in serum.

### 3.2. Anti-Inflammatory Effect of taVNS on Spleen Tissue of LPS-Induced Inflammation

Since the concept of CAP from the Tracey group in 2000, this theory has been confirmed in multilevel organs such as the spleen, lung, and gut [5]. These organs are regulated by efferent fibers of the VN when inflammation is evoked, following which the vago-parasympathetic reflex activates and then targets these multi-organs of α7nAChRs for anti-inflammation [6,23,30]. Therefore, we investigated whether taVNS stimulated the vagal anti-inflammatory effect of the spleen, lung, and intestine. The swelling of LPS-induced spleens was relieved by treatment of taVNS as the LPS-untreated groups (Sham) (Figure 3A). The rate of MPO expression of the spleen in LPS-induced endotoxemia increased (four to five-fold) compared to the sham groups. Intensive expression with MPO on the spleens of the LPS-induced inflammation indicated an increase in leukocyte extravasation to the spleen during the inflammatory condition. However, it significantly reduced the expression levels of MPO on the spleen of LPS + taVNS treated mice in a pulse frequency-dependent manner (Figure 3B). The qRT-PCR analysis showed that taVNS decreased the relative expression of genes such as *TNF-a*, *IL-1β*, *IL-6*, and *IL-8*, which have a stimulatory role in inflammation (Figure 3C), compared to the only LPS-treated group. The expression levels of these pro-inflammatory cytokine genes were significantly decreased when taVNS treatment was applied to LPS-induced endotoxemia regardless of the taVNS parameter. Compared to the taVNS-treated groups with different parameters, taVNS treatment with 15 Hz displayed a more significant decrease in the mRNA levels of *TNF-a*, *IL-1β*, *IL-6*, and *IL-8* than taVNS treatment with 25 Hz. The taVNS with 15 Hz_5 min treatment group showed lower expression levels of *TNF-a* and *IL-6* than the taVNS with 15 Hz_10 min treatment group, but not the taVNS with 25 Hz treatment group. The relative expression of anti-inflammatory cytokine mRNA, including *IL-10* and *TGF-β* encoding genes, was also determined using different pulse frequencies of taVNS. The significantly decreased *IL-10* and *TGF-β* gene expressions were evaluated in the taVNS-treated groups, except for taVNS with 25 Hz_10 min in *TGF-β* gene expression. These results demonstrated that taVNS significantly reduced the inflammatory reaction of the spleen in LPS-induced endotoxemia, and the pulse frequency of taVNS is capable of regulating the expression levels of inflammatory cytokine genes in the spleen.

### 3.3. Anti-Inflammatory Effect of taVNS on Lung Injury of LPS-Induced Endotoxemia

Next, we validated whether taVNS reversed the inflammatory response of the lung in LPS-induced endotoxemia. The representative images of the lung after taVNS treatment on the LPS endotoxemia model show the morphological difference between the LPS and LPS with taVNS groups (upper panel of Figure 4A). H&E staining also determined a significant anti-inflammatory effect of taVNS on LPS-induced damage in mice lungs, compared to only LPS-exposed mice (lower panel of Figure 4A, Appendix A). The congested alveolar wall was distinguishable, and the edema phenomenon of the interalveolar septum was decreased in the LPS + taVNS group, compared to the Sham and LPS-treated group. Next, MPO stain was used in this experiment to evaluate inflammatory cell infiltration. As shown in Figure 4B, positive staining of MPO was highly observed in the lung of LPS-induced endotoxemia, but upon treatment with taVNS, there was a remarkable reduction in the strong expression of MPO of lung injury by LPS. The proportion of MPO-positive stained area significantly decreased and disappeared in taVNS-treated groups compared to only LPS-induced group (Figure 3B). The lung injury score and coverage rate of MPO stain in the lung are presented in Figure 4C. The relative expression levels of pro- and anti-inflammatory cytokines were also analyzed in treatment with or without taVNS on LPS-induced lung injury. Consistent with the qRT-PCR spleen results, all groups with taVNS showed significant alleviation of pro-inflammatory cytokine expression in lungs after taVNS treatment. Above all, treatment of taVNS with 15 Hz caused a significant decrease in the expression of *IL-1β* and *IL-6* encoding genes compared with 25 Hz, but the fold change increase or decrease in the taVNS-treated group with different time parameters (5 min and 10 min) was not observed in the same parameter of taVNS (Figure 4D). These results showed the protection and recovery effects of taVNS on LPS-induced lung injury.

### 3.4. Anti-Inflammatory Effect of taVNS on Intestinal Inflammation Induced by LPS

Regarding the anti-inflammatory effect of taVNS on the intestine, the pro-inflammatory cytokine levels of intestines were determined by ELISA and indicated a significant downregulation of TNF-α, IL-6, and IL-1β compared with the only LPS-treated group (Figure 5A). Different expression levels of these pro-inflammatory cytokines were observed in the taVNS with 15 Hz and 25 Hz groups (decrease rate in the 15 Hz_10 min treated group, −65% (TNF-α), −85% (IL-6), −38% (IL-1β); 25 Hz_10 min group, 22% (TNF-α), −10% (IL-6), −15% (IL-1β). Particularly, taVNS with 15 Hz showed some variation compared to 25 Hz. This may imply that the stimulation condition of the taVNS with 15 Hz group was unstable.

The intestine in LPS-induced endotoxemia was swollen and longer compared to the untreated group. The treatment of taVNS on LPS-induced inflammation reduced these morphological changes in the LPS-treated group (Figure 5B). To assess the protective capacity of taVNS on intestine injury via LPS, histological evaluation was performed using H&E and MPO stain in LPS-induced intestine with taVNS 15 Hz and/or 25 Hz (Figure 5C). Histological scores and MPO intensity-graph of the intestine showed the effective anti-inflammatory capacity of taVNS. Moreover, a difference in the recovery efficacy of taVNS between 15 Hz and 25 Hz on acute inflammation was also observed in mice gut.

### 3.5. Electrical Frequency of the taVNS Parameter Regulated Inflammatory Cytokines in the Serum of LPS-Induced Endotoxemia

We next investigated the whole inflammatory molecule level changes triggered at two different pulse frequency parameters upon taVNS treatment of the LPS-induced endotoxemia to analyze the different anti-inflammatory effects of taVNS on systemic inflammation. The heatmap image shows the differentially expressed 50 cytokines/chemokines related to inflammation and immune response between LPS-induced inflammation treated with or without taVNS. taVNS downregulated most of the cytokines/chemokines activated by LPS induction (Figure 6A). taVNS with 15 Hz suppressed the cytokines/chemokines evaluated values more severely than taVNS with 25 Hz. The scatter plot analysis also presents the differential downregulation of the upregulated cytokines/chemokines by LPS between taVNS with 15 Hz and 25 Hz, as shown by the distribution of cytokines/chemokines dots (Figure 6B). Of these 50 cytokines, the pixel intensity of TNF-α, IL-6, IL-1β, TGF-β1, TLR4, and IL-10 and fold change of chemokines (CCL/CXCL) and interleukins (IL) presented in Figure 6C and Appendix A also indicate differential modulation of inflammatory cytokine levels through different taVNS pulse frequencies. The pixel intensity of 34 significantly downregulated chemokine/cytokines at 15 Hz compared to 25 Hz is presented in Appendix A. These results imply that taVNS differentially improved systemic inflammation via modulation of the expression levels of cytokines/chemokines using changes in the pulse frequency parameter of taVNS.

## 4. Discussion

VNS has three main fibers: A-, B-, and C-fibers, which can be delivered by sensory afferent and motor efferent signals to regulate vital functions in the body’s autonomic nervous system [31]. These nerve fibers of the VN innervate different physiological changes via combined or individual fiber form [32]. For example, the activation of A- and B-fibers of the VN is associated with anti-inflammatory effects, while C-fibers activation is involved in triggering cardioinhibitory effects [24,33]. Each fiber of the VN has distinct stimulation thresholds for activation because of its different axon diameters, conduction velocity, and myelination [34,35]. Typically, the higher the stimulation current levels, the smaller nerve fiber activation in the peripheral nervous system occurs [36]. The increasing pulse width and amplitude also selectively activate small diameter nerve fibers while inhibiting the activation of large diameter fibers [37]. This means that the bioelectric stimulation parameter can differentially modulate the activation of VN fibers and consequently exhibit desired physiological effects [32]. The most important goal of taVNS research is the selective and efficient nerve activation using taVNS parameters, such as pulse frequency, duration, amplitude, time, and electrical current rate, to apply as a therapeutic tool. However, it has been poorly defined, and the stimulation parameter of taVNS for neuromodulation therapy needs to be optimized. We first tested how different pulse frequencies and times of taVNS affect the anti-inflammatory effects on VN innervated tissues and serum using an LPS-induced endotoxemia model. taVNS effectively reduced LPS-induced inflammation, as indicated by a decrease in pro-inflammatory cytokines expression, histopathological scores, and leukocyte infiltration. These anti-inflammatory effects of taVNS were changed by the stimulation parameter of pulse frequency and time. Among them, the result of the cytokine array showed the most obvious difference between 15 Hz and 25 Hz of taVNS during inhibition of the LPS-induced inflammation. taVNS with 15 Hz severely downregulated the levels of cytokines and chemokines in serum, whereas taVNS with 25 Hz did not. These findings indicate that the rate of inflammatory cytokine production can be modulated by regulating the pulse frequency of taVNS in various inflammatory conditions.

In 2020, Piruzyan et al. showed that electrical stimulation with a high-frequency pulse current more effectively suppressed the excessive production of inflammatory cytokines than a low-frequency pulse current [38]. However, this report included different electrical stimulation systems and anti-inflammatory mechanisms without stimulation of the VN. Another recent study supports our results that low pulsing frequency selectively provided the optimal intensity range to activate the A- and B-fibers of the VN [39]. The low stimulation threshold of A- and B-VN fibers need to achieve the activating vagal anti-inflammatory pathway that distinguishes the C-fiber of the VN to regulate the heart rate [24]. The electrical stimulation condition of taVNS with 15 Hz may be more effective in the activation of CAP through A- and B-fibers of vagal signaling than a higher pulse frequency of taVNS. This does not mean that taVNS with 15 Hz is optimal for regulating inflammation because proper inflammation response is necessary to defend the body against infection. Tsaava et al. suggested that different pulse widths, duration, and amplitude may play an important role in the modulation of inflammatory-related cytokines via the VN [29]. However, our taVNS equipment could not regulate other parameters except Hz and time, which was a limitation of this study. The taVNS parameters need to be fine-tuned in the future.

The fibers in the cervical VN consist of a mixed formation, which co-activates side effect-inducing fibers [40]. It has been known that the inhibitory effect of intestinal inflammation was supposed to be related to vagal C-fibers with a high stimulation threshold but typically also activated other fibers in the cervical VN [41,42]. The selective activation of fibers in the VN is essential to the treatment of distinct diseases if VNS is used as a treatment method [43]. Compared with cervical VNS, taVNS is easy to apply for therapy on ears. In addition to convenience, taVNS is a safe method because it indirectly regulates the VN without directly connecting with the vagal fibers. Thus, taVNS has not been reported to cause cardiac dysfunction [44].

Inflammation is a protective reaction of the host against exogenous pathogens, stress, and injury that must control and balance the body’s immune system [43]. Excessive inflammatory response due to autoimmunity or uncontrolled inflammatory pathway of host cells leads to several inflammatory diseases such as rheumatoid arthritis, atopic dermatitis, and chronic inflammation in humans [45,46]. COVID-19 also causes serious inflammatory responses and many chemical drug therapies have been trialed such as corticosteroids, tocilizumab, IL-6 inhibitor, and intravenous immunoglobulin. However, specific approaches for COVID-19 are currently lacking [16,47,48,49,50,51]. The anti-inflammatory role of VNS can specifically reduce overproduced inflammatory cytokine levels via CAP activation. Two case studies of using VNS treatment for COVID-19 have highlighted its potential clinical benefit in treating patients with COVID-19 [21,52]. taVNS is a non-invasive and safe therapy. Therefore, VNS may be considered as a supplement treatment if large clinical trials prove its efficacy in treating patients with COVID-19.

## 5. Conclusions

We developed a stimulating electrode system of transcutaneous auricular VN and determined the effective anti-inflammation capacity of taVNS. Different taVNS pulse frequency parameters differentially modulated the rate of inflammation injuries on the spleen, lung, and gut, regulation of inflammatory-related cytokines expression, and chemokines levels. taVNS appears to be an effective therapeutic tool against inflammation disorders in humans following optimization of the taVNS parameters.

## Figures and Tables

**Figure 1 biomedicines-10-00247-f001:**
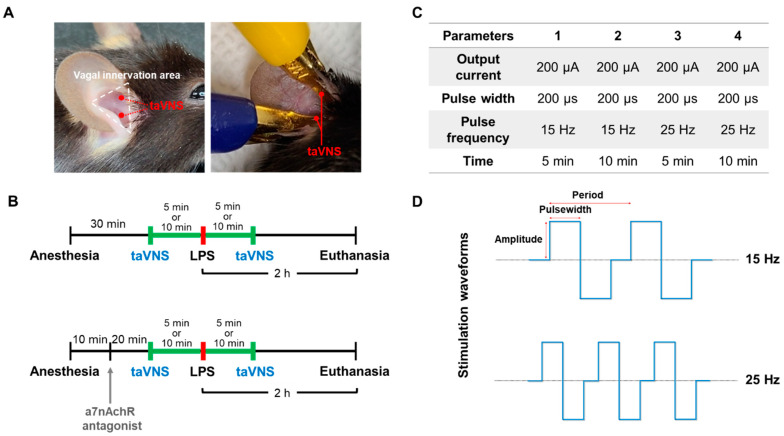
Experimental design and procedure of taVNS in the LPS-induced endotoxemia in mice. (**A**) Mice were bilaterally treated with taVNS, followed by the cymba and cavum concha of the vagus innervation area in the ear. (**B**) Under anesthesia, mice were stimulated with taVNS treatments for 5 or 10 min before and after LPS injection. The animals were euthanized after 2 h of LPS administration, and the whole blood and tissues were collected. (**C**) Four different taVNS parameters were used during stimulation in this study. (**D**) Schematic image showing the charge-balanced stimulation waveform of taVNS with low pulse frequency (**top**) and high pulse frequency (**bottom**).

**Figure 2 biomedicines-10-00247-f002:**
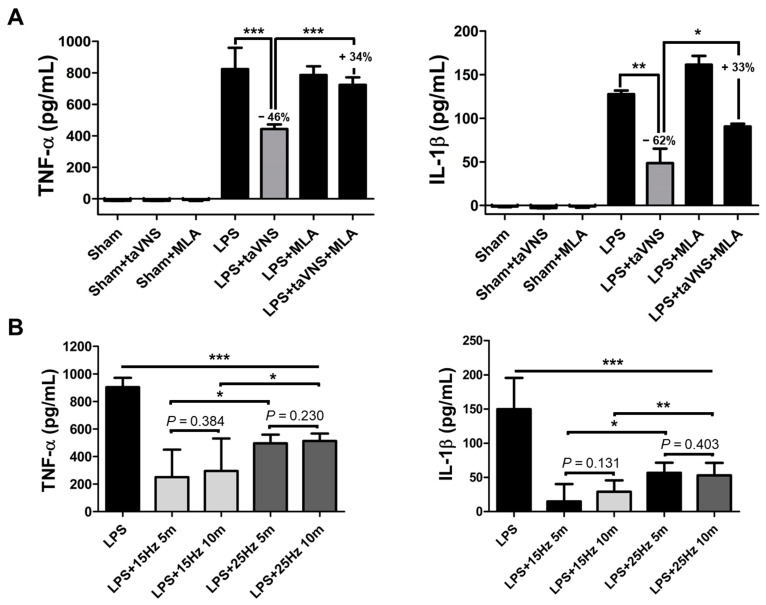
A decrease in pro-inflammatory cytokine releases by taVNS in LPS-induced endotoxemia. (**A**) Mice were treated with taVNS and/or MLA treatment for acute inflammation from LPS. The expression levels of serum TNF-α and IL-1β were measured by ELISA. The decrease and recovery percentages of each cytokine were represented as a number. (**B**) The expression levels of serum TNF-α and IL-1β were determined using ELISA pre- and post-treatment of taVNS with different pulse frequency and time parameters. Data have been presented as the means and SD; * *p* < 0.05, ** *p* < 0.01, and *** *p* < 0.001, compared to the corresponding control.

**Figure 3 biomedicines-10-00247-f003:**
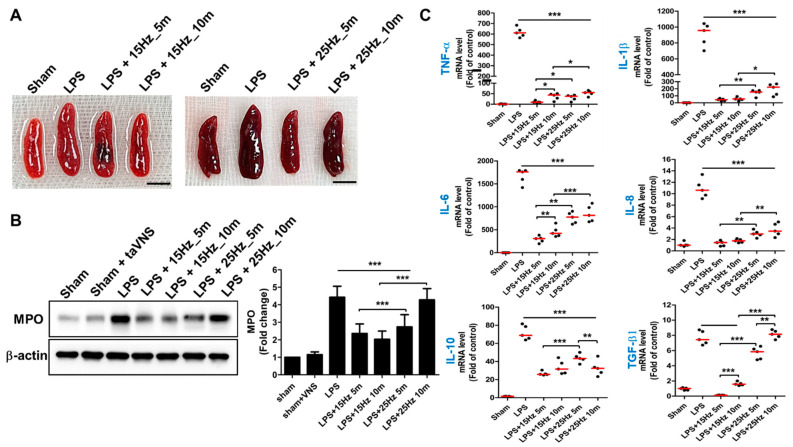
taVNS reduced inflammation of the spleen in LPS-induced endotoxemia. (**A**) Representative images of the spleen after treatment of taVNS with different pulse frequencies and time parameters. Scale bar: 5 mm. (**B**) Western blot analysis using an antibody against MPO was used to analyze the relative expression levels of neutrophils in the indicated stimulation conditions of taVNS on LPS-induced inflammation. The intensity ratios for MPO were presented as a graph using ImageJ. (**C**) The mRNA levels of pro- and anti-inflammatory cytokines were determined by qPCR. All results have been presented as the means and SD; * *p* < 0.05, ** *p* < 0.01, and *** *p* < 0.001, compared to the control.

**Figure 4 biomedicines-10-00247-f004:**
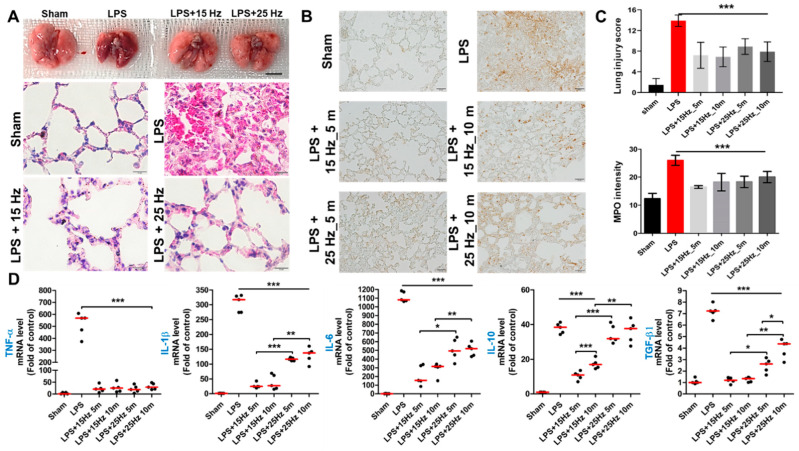
Anti-inflammatory effect of taVNS on the lung of LPS-induced endotoxemia. (**A**) The representative images of the lung after treatment of taVNS with 15 Hz and 25 Hz. Scale bar: 5 mm. The results of H&E staining were determined using the lung injury score upon observation under a light microscope. Scale bar: 10 μm. (**B**) MPO staining was determined by the infiltrated immune cells on the lung. Scale bar: 20 μm. (**C**) The relative scoring of lung injuries was compared as a graph in three independent images. Positively stained area was measured in three different images in each group and then represented on a graph. (**D**) The expression levels of pro- and anti-inflammatory cytokines genes were determined by qPCR, and significance was compared among groups. Data have been presented as the means and standard deviation (n = 5–10); * *p* < 0.05, ** *p* < 0.01, and *** *p* < 0.001, as compared to the corresponding control.

**Figure 5 biomedicines-10-00247-f005:**
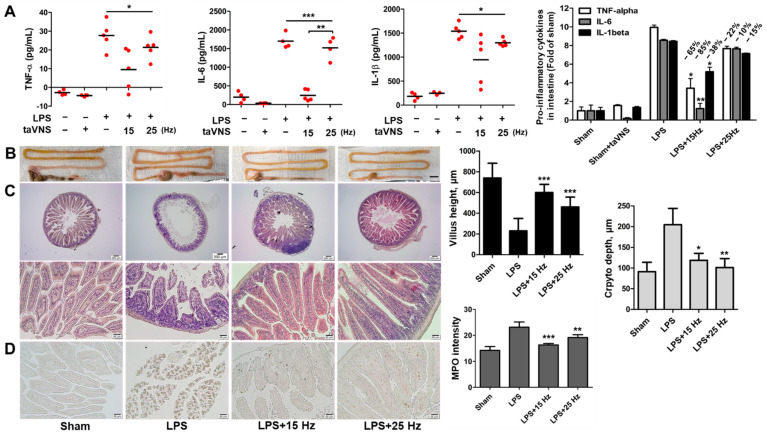
taVNS induced anti-inflammatory effect on the intestine of LPS-induced endotoxemia. (**A**) Determination of the pro-inflammatory cytokines in the intestine of LPS-induced inflammation with or without taVNS treatment. The relative expression levels of pro-inflammatory cytokines in response to taVNS treatment of LPS-induced endotoxemia were indicated. Fold change of groups was represented as a graph and calculated decreases rate in taVNS treated groups. (**B**) The morphological changes of the intestine were represented. Scale bar: 10 mm. (**C**) H&E staining showed the histological change of the intestine upon observation under a light microscope. Scale bar: 200 μm (upper panels) and 50 μm (lower panels). The villus height and crypto depth of the intestine were determined and represented by a graph. (**D**) Immunohistochemical staining with anti-MPO antibodies showed a significant decrease in neutrophils in LPS-induced inflammation by interventions as of taVNS. Scale bar: 50 μm. All taVNS groups were treated with electrical stimulation for 10 min. Data have been presented as the means and SD; * *p* < 0.05, ** *p* < 0.01, and *** *p* < 0.001, compared to the corresponding control.

**Figure 6 biomedicines-10-00247-f006:**
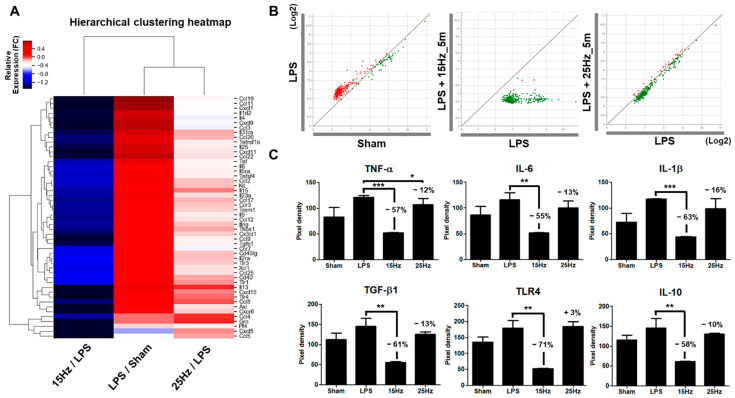
Changes in inflammatory cytokines and chemokines in the plasma of LPS-induced endotoxemia with taVNS. (**A**) Most cytokines and chemokines were upregulated by LPS injection (LPS/Sham). Both taVNS groups (15 Hz and 25 Hz) attenuated the expression levels of cytokines and chemokines in serum. taVNS with 15 Hz was more significantly reduced than with 25 Hz. Clustering software generated a heatmap. The color depicts the levels of fold change. (**B**) Scatter plot analysis showed the expression changes of cytokines and chemokines for the LPS-induced inflammation verse sham (**left**), taVNS with 15 Hz on the LPS-induced inflammation versus the LPS-induced inflammation (**middle**), and taVNS with 25 Hz on the LPS-induced inflammation versus the LPS-induced inflammation (**right**). Upregulation is presented as red dots and downregulation as green dots. (**C**) The representative pixel density of pro- and anti-inflammatory-related cytokines, namely TNF-α, IL-6, IL-1β, TGF-β1, TLR4, and IL-10 in the serum of the sham, LPS applied groups and taVNS with 15 Hz or 25 Hz on LPS-induced groups. Data have been presented as the means and SD; * *p* < 0.05, ** *p* < 0.01, and *** *p* < 0.001 to determine the significance levels compared to the LPS group.

## Data Availability

Data are all contained within the article.

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
