# Peer review of "Different Transcutaneous Auricular Vagus Nerve Stimulation Parameters Modulate the Anti-Inflammatory Effects on Lipopolysaccharide-Induced Acute Inflammation in Mice"

_biomedicines, 2022, doi:10.3390/biomedicines10020247_

Round 1

Reviewer 1 Report

Overview

In the current study by Go et al., the researchers use transcutaneous auricular vagus nerve stimulation to alter cytokine expression in blood, spleen, lung and intestine upon LPS-induced systemic inflammation in mice.

Overall comment/ major concerns

Introduction:

The background to the study is well described and gives an overview of the state of the art and the reasoning and application for this study.

Methods:

The descriptions in general, but especially of LPS administration and taVNS treatments are hard to understand and inconsistent between text and figure legend. This has to be rewritten and clarified.

Why/ how was the time point of 2h chosen for these experiments? Can the authors explain why in some samples, cytokines were measured by ELISA, in others on mRNA level by qPCR?

In my experience, 1-2h after LPS administration is the time when in many tissues the highest change in mRNA transcripts of cytokines can be detected. On protein levels, cytokines often peak at later time points. Have there been any time-course experiments done as an initial experiment? Is there any reference that points to this time point for protein measurement? Is there any prove that this particular time point can represent the full extent of cytokine response for each setting? If not, at least a second time point would be interesting. Is it possible that there is only a time shift in the cytokine response? Is there any information on the overall outcome, the resolution of inflammation, available?

The supplementary material cannot be opened/ viewed by the reviewer (file damage or similar).

Results:

Chapter 3.1 is clearly a method description and is in the wrong place under the results section.

Discussion:

The discussion is clear and understandable, nevertheless, a comment on limitations of the study should be included.

Specific comments:

Line 18f systematic: do you mean systemic?

Line 84ff: “a single ip injection of LPS was conducted between 5 or 10 min before the stimulation with taVNS.” This sentence doesn’t make sense. Refer to Figure 1B, where the experiment is depicted and rewrite sentence.

Line 86ff: “Animals were sacrificed within 2h after LPS administration.“ At what timepoint exactly? “Within” could be 10 min, but also 1h and 59 min, which makes a huge difference in terms of cytokine stimulation. According to Figure 1B and the figure legend, animals were sacrificed 2h after taVNS treatment, which seems to be 2h and 5 minutes after LPS administration, but the text has to match the figure description.

Line 94: “taVNS was applied twice for 5 or 10 min between LPS administration” there is only one LPS administration, so the LPS was applied between two taVNS treatments, right? Please correct the text accordingly.

Line 110: “The concentrations of TNF concentrations of TNF-fusing mouse-specific ELISA kits…” This is not a full sentence and doesn’t make sense.

Line 201ff and many others: “LPS-induced mice” Please use LPS-induced endotoxemia or LPS-induced inflammation, but not LPS-induced mice! Rephrase in whole manuscript

Reviewer 2 Report

The manuscript entitled, "Different transcutaneous auricular vagus nerve stimulation parameters modulate the anti-inflammatory effects on lipopolysaccharide-induced acute inflammation in mice", is well designed and stands as an interesting addition to the field. However, in discussion part the authors tried to justify that the VNS stimulation may be appropriate in treating the inflammation due to COVID-19. 

I would ask author to remove those lines from lines 428 to 436 and replace it with relevant discussion. 

Round 2

Reviewer 1 Report

Broad comments:

The authors haven’t answered my question, why in some samples, cytokines were measured by ELISA, in others on mRNA level by qPCR?

In serum, it makes sense to use ELISA and measure protein levels. But why was in spleen and lung qPCR used, while for the intestine ELISA was used for the detection of cytokines? Where qPCR was used, can the same effects found on mRNA level be seen on protein level as well?

Specific comments:

Line 101: there is still a confusing description: “taVNS was applied ….. for 5 or 10 min between LPS injection.” Please correct

Supplementary Figure 1:

Why is the section of the LPS treated tissue in a different magnification than the others (different scale bar)?
